

# Nested Markov chain hyper-heuristic (NMHH): a hybrid hyper-heuristic framework for single-objective continuous problems

Nándor Bándi and Noémi Gaskó

Faculty of Mathematics and Computer Science, Babeş-Bolyai University, Cluj-Napoca, Romania

## ABSTRACT

This article introduces a new hybrid hyper-heuristic framework that deals with single-objective continuous optimization problems. This approach employs a nested Markov chain on the base level in the search for the best-performing operators and their sequences and simulated annealing on the hyperlevel, which evolves the chain and the operator parameters. The novelty of the approach consists of the upper level of the Markov chain expressing the hybridization of global and local search operators and the lower level automatically selecting the best-performing operator sequences for the problem. Numerical experiments conducted on well-known benchmark functions and the comparison with another hyper-heuristic framework and six state-of-the-art metaheuristics show the effectiveness of the proposed approach.

# INTRODUCTION

Optimization is an essential task not only in computer science but also in other research fields. Most processes can be described as optimization problems, where the best solution needs to be found from the set of all feasible solutions.

The literature contains many optimization algorithms and heuristics, some inspired by nature, others inspired by physics, iterative, and hybrid. Finding the appropriate approach is often problem-specific and can be tedious. Population-based methods may approximate the global optimum but at a high computational cost. Iterative methods converge to a local minimum faster but are highly dependent on the initial solution.

Recently, hyper-heuristic algorithms have been of huge interest, as they provide an automatic way of selecting or generating heuristics for unseen problems (see *Ryser-Welch & Miller, 2014* for a review). These approaches can be thought of as the optimization of the optimization process. The selection or generation of heuristics yields a problem-specific optimization algorithm that in many cases performs better than a single standard heuristic.

Application possibilities where different hyper-heuristics were used include timetabling (*Burke, Qu & Soghier, 2014*; *Burke, Silva & Soubeiga, 2005*; *Pillay, 2012*), the vehicle routing problem (*Qin et al., 2021*; *Olgun, Koç & Altıparmak, 2021*), and scheduling

Corresponding author
Noémi Gaskó,
gaskonomi@cs.ubbcluj.ro

problems (*Salhi & Vázquez Rodríguez, 2014*), aircraft structural design (*Allen, Coates & Trevelyan, 2013*).

Although several hyper-heuristic frameworks have been proposed, most of them are concerned with specific combinatorial optimization problems; only a few are designed to solve continuous numerical optimization problems. A research gap exists regarding hyper-heuristic frameworks that balance the exploration-exploitation rate and tune the operator parameters in an online fashion. As another research gap we can mention the lack of generality of the proposed hyper-heuristic frameworks, the majority of them are incorporating domain specific knowledge about a specific problem (for example *Guerriero & Saccomanno, 2023*).

The goal of this study is to propose a new hyper-heuristic framework and to present its advantages for single-objective continuous problems and the comparison with a well-known hyper-heuristic and six state-of-the-art metaheuristics. The novelty of our approach consists of introducing a nested Markov chain to the base level for the search for the best-performing heuristic operators and their sequences. Simulated annealing is used on the hyperlevel, which evolves the chain and the operator parameters. In our approach, the upper level of the Markov chain expressing the hybridization of global and local search operators and the lower level automatically selecting the best-performing operator sequences for the problem. The general formulation of the model allows the usage of other arbitrary operators, as well. Our model can be used to achieve good optimization results without the user having deep domain (problem specific) knowledge. The limitations of our approach are similar to other hyper-heuristics, finding the right operator configurations and balance can require many function evaluations.

The remainder of the article is organized as follows: the "Related Work" section describes the related work, the "Proposed Model" section presents the proposed framework, and the "Numerical Experiments" section describes the numerical experiments conducted. The article ends with conclusions and further research directions.

## RELATED WORK

The literature proposes several hyper-heuristic classifications. Two main categories appear in *Burke et al. (2010)*: selection-based and generation-based. Selection-based approaches pick the best-performing heuristics from an existing catalogue, while generation-based approaches design new algorithms from existing components and create problem-specific ones. Four categories of heuristic selection are mentioned in *Chakhlevitch & Cowling (2008)*. Metaheuristic-based approaches employ genetic algorithms (*Cowling, Kendall & Soubeiga, 2001*), simulated annealing (*Bai & Kendall, 2005*), tabu search (*Kendall & Hussin, 2005*) or some other metaheuristic for the selection process. In *Bándi & Gaskó (2023)* on the hyperlevel, a simulated annealing algorithm is used, and on the base level a genetic algorithm, a differential evolution algorithm and a grey wolf optimizer. Random approaches employ uniform selection (*Cowling & Chakhlevitch, 2003*). Other approaches use reinforcement learning for adaptive selection. *McClymont & Keedwell (2011)* adapts a Markov chain that models heuristic sequences. *Karapetyan, Punnen & Parkes (2017)*

uses the Conditional Markov Chain Search (CMCS) algorithm for the bipartite Boolean quadratic programming problem (BBQP). Greedy selection methods preliminarily evaluate all heuristics and choose the one that performs best at each step (*Cowling, Kendall & Soubeiga, 2001*). *Oteiza, Ardenghi & Brignole (2021)* presents a parallel cooperative hyper-heuristic optimizer (PCHO), which is used to solve systems of nonlinear algebraic equations (with equality and inequality constraints). It uses a master-worker architecture with three algorithms on the worker level: GA, SA and PSO.

Since our proposed hyper-heuristic framework uses a hybridization of global and local search, we will present existing approaches in this category.

The use of local search algorithms is a straightforward direction in the study of hyper-heuristics but was used mainly for combinatorial optimization problems. *Burke, Kendall & Soubeiga (2003)* incorporates tabu search in hyper-heuristics for the timetabling problem. *Turky et al. (2020)* proposes a two-stage hyper-heuristic to control the local search and its operators; the framework is used for two combinatorial optimization problems. *Hsiao, Chiang & Fu (2012)* proposes a hyper-heuristic based on variable neighbourhood search, where local search is used and tested for four combinatorial optimization problems. *Soria-Alcaraz et al. (2016)* designs a hyper-heuristic based on an iterated local search algorithm for a course timetabling problem. Additionally, the reviews (*Ryser-Welch & Miller, 2014*; *Drake et al., 2020*) present several hyper-heuristic frameworks, such as HyFlex (hyper-heuristics flexible framework) for combinatorial optimization problems (*Ochoa et al., 2012*), or Hyperion (*Swan, Özcan & Kendall, 2011*) for the Boolean satisfiability problem.

In terms of continuous optimization, *Oliva et al. (2022)* proposes the HHBNO framework, a hyper-heuristic approach based on Bayesian learning for single-objective continuous problems. The framework evolves heuristic sequences by learning their interdependencies and estimates the best-performing heuristic distributions. In *Tapia-Avitia et al. (2022)*, an artificial neural network is trained to identify patterns that can be used to learn the best-performing heuristic sequences. *Cruz-Duarte et al. (2021)* proposes a new framework for continuous optimization problems, where new sequences are designed with the help of different search operators.

Our proposed hyper-heuristic framework incorporates local and global search algorithms, which were mostly used for combinatorial optimization problems before. Another advantage of the proposed method consists in the general structure of the base level, which can be easily extended with other algorithms. At the same time the framework preserves the general nature, no domain specific knowledge is needed for the optimisation process.

## PROPOSED MODEL

The structure of the proposed hyper-heuristic framework is presented in Fig. 1. The approach is based on two levels. The base level optimizes the problem, starting with a population of candidate solutions and a limited number of function evaluations for each candidate. The hyperlevel guides and improves the base level. The hyperlevel searches

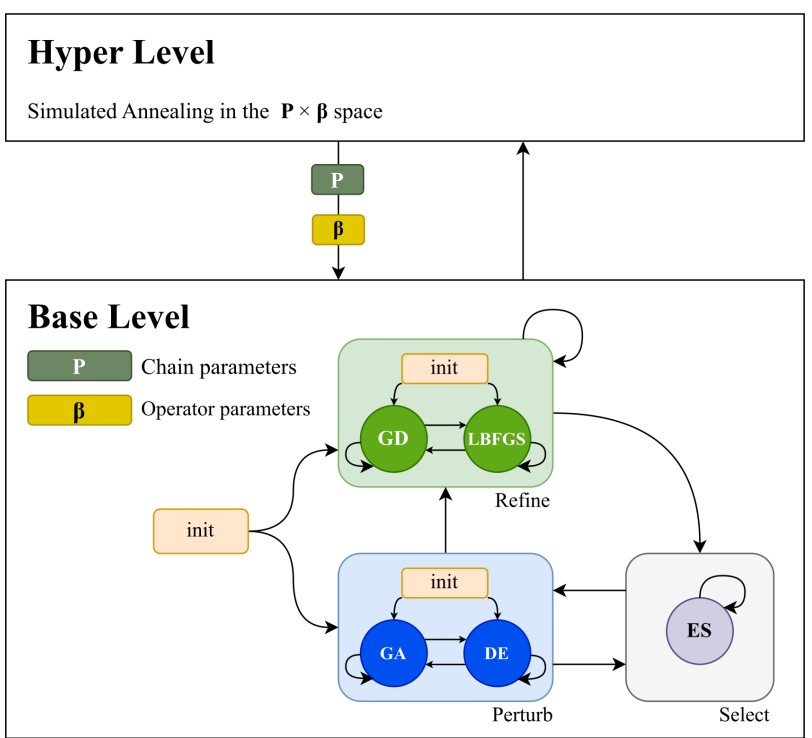

**Figure 1** **The structure of the hyper-heuristic framework.** The base level is used to optimize the problem using a given number of function evaluations. On this level a genetic algorithm (GA) and differential evolution (DE) operator is used in the perturb category, an elitist selector (ES) in the select category, and in the refiner category the gradient descent (GD) and limited-memory Broyden, Fletcher, Goldfarb, Shanno algorithm (LBFGS) operators are used. The hyper level is used to guide and improve the base level.

the algorithm space by finding the best-performing operator sequences and operator parameters *via* simulated annealing. The base level performs the optimization according to the operator sequence modelled by a nested Markov chain. The first layer models the transitions between perturb, selection, and refinement operators, and the second layer models the operator sequences of each category.

Our approach contains a genetic algorithm (GA) and a differential evolution (DE) operator in the perturb category and an elitist selector (ES) in the select category. It incorporates a refiner category containing the gradient descent (GD) and Limited-memory Broyden–Fletcher–Goldfarb–Shanno algorithm (LBFGS) operators that perform the local search. In this way, the base level is parameterized so that it can express a continuum between exploration and exploitation.

*Formal definition.* A more formal definition of the parameterization of the operator sequence that the base level applies during optimization can be given in the following way. Let $\mathcal{C}$ denote the set of operator categories in the base level, and $c \in \mathcal{C}$ an operator category.

Let

$$c_1, c_2 \ldots c_i \sim \pi_{\mathcal{C}}, P_{\mathcal{C}}, o_1^c, o_2^c \ldots o_j^c \sim \pi_c, P_c, \forall c \in \mathcal{C}$$

denote the sequence of categories $c_i$ modeled by the Markov chain parameterized by the initial distribution and transition matrix $\pi_{\mathcal{C}}, P_{\mathcal{C}}$ and the sequence of operators $o_j^c$ modeled by the Markov chain associated to category $c$ parameterized by the initial distribution and transition matrix $\pi_c, P_c$.

Then the sequence

$$o_1^{c_1}, o_2^{c_2} \ldots o_i^{c_i} \sim \pi_{\mathcal{C}}, P_{\mathcal{C}}, \pi_c, P_c \, \forall c \in \mathcal{C}$$

is called the operator sequence of the base level modeled by the nested Markov chain.

The set of all operator parameters associated to operators in category $c$ is denoted by $\beta_c$. The hyper level searches in the sequence and parameter space

$$P \times \beta = \pi_{\mathcal{C}} \times P_{\mathcal{C}} \times \prod_{c_i \in \mathcal{C}} \pi_{c_i} \times P_{c_i} \times \beta_{c_i}$$

*via* simulated annealing using linear multiplicative cooling for the best performing base level configuration.

*Hyper-heuristic optimization algorithm.* At each step, a statistically significant number of base-level evaluations are performed and the performance metric is the median plus interquartile range of the costs. The next step in the design space is taken by perturbing the previous point by a normally distributed noise factor scaled according to the parameter bounds. The advantage of this formalization lies in the expressiveness of the base level as it allows the selection and combination of sets of operators that have different roles (exploration, exploitation, selection).

The model can perform well in high-dimensional settings, as it can iteratively find the equilibria between exploration and exploitation operators. The random initialization of the hyper-heuristic is presented in Algorithm 1. The hyper-heuristic search process using simulated annealing and the determination of the next simulated annealing step are detailed in Algorithm 2 and Algorithm 3. Algorithm 4 shows the base-level optimization procedure that is modelled by the Markov chain and operator parameters.

*Operators used.* The GA operator is parameterized to allow it to express both arithmetic and one-point crossover; the mutation is carried out by adding a normally distributed noise factor as detailed in Algorithm 5. The DE/rand/1/bin scheme is used for the differential evolution operator; it is parameterized by the crossover rate and scaling factor. The local search operators are parameterized by the initial step size and the number of iterations performed. The LBFGS operator also exposes the $c_1, c_2$ parameters that control the step length in the line search phase. The selection operator uses the elitist strategy.

*Population evolution.* All operators in the perturb category generate a new population of candidates. The operators in the refine category perform an iterated local search starting with these candidate solutions. The elitist selection operator then selects the best-performing points from the old and new populations to become the next generation. All perturbed points landing in the attraction basin of a better solution are selected into the next generation when the refiner operators iterate the points closer to these attractors and the ES operator selects them.

---

**Algorithm 1** Hyper-heuristic parameter initialization

---

$\{\, l_\theta, u_\theta$ - bounds of parameter $\theta\, \}$

$P_\mathcal{C} \leftarrow U(T)$ {// random uniform transition matrix}

$\pi_\mathcal{C} \leftarrow U(\pi)$ {// random uniform initial distribution}

**for all** $\theta \in \beta_c, \forall c \in \mathcal{C}$ **do**

   $\theta \leftarrow U(l_\theta, u_\theta)$ {// random uniform parameter within bounds}

**end for**

**for all** $c \in \mathcal{C}$ **do**

   $P_c \leftarrow U(T)$ {// random uniform transition matrix}

   $\pi_c \leftarrow U(\pi)$ {// random uniform initial distribution}

**end for**

---

**Algorithm 2** Optimization in the hyper level

---

$\{\, h_l$ - hyperheuristic step limit$\}$

$\{\, h_f$ - function evaluation limit of offspring per step$\}$

$\{\, T$ - initial simulated annealing temperature$\}$

$\{\, \alpha$ - linear multiplicative cooling coefficient$\}$

$\{\, h_p$ - base level performance sample size$\}$

initialize chain and operator parameters (algorithm 1)

$t \leftarrow T$

$f_{best} \leftarrow \infty$

**for** $l \leftarrow 0; l < h_l; l \leftarrow l + 1$ **do**

   $P', \beta' \leftarrow$ mutation of $P, \beta$ (algorithm 3)

   **for** $s \leftarrow 0; s < h_p; s \leftarrow s + 1$ **do**

      $p_{l,s} \leftarrow$ performance sample for $P', \beta'$ (algorithm 4)

   **end for**

   $f \leftarrow IQR(p_l) + median(p_l)$

   **if** $f < f_{best}$ **then**

      $P, \beta \leftarrow P', \beta'$

      $f_{best} \leftarrow f$

   **else**

      **if** $U(0,1) < e^{\frac{f_{best} - f}{t}}$ **then**

         $P, \beta \leftarrow P', \beta'$

         $f_{best} \leftarrow f$

      **end if**

   **end if**

   $t \leftarrow \frac{T}{1 + \alpha \cdot l}$

**end for**

---

---

**Algorithm 3** Parameter mutation

> **for all** $c \in \mathcal{C}$ **do**
>> $\beta'_c \leftarrow \beta_c$
> **end for**
> **for all** $\theta' \in \beta'_c, \forall c \in \mathcal{C}$ **do**
>> { // perturb parameters keeping them in bounds }
>> $\epsilon \sim \mathcal{N}(0, \frac{1}{3})$
>> $\theta' \leftarrow \theta' + \epsilon(u_{\theta'} - l_{\theta'})$
>> $\theta' \leftarrow \max(l_{\theta'}, \theta')$
>> $\theta' \leftarrow \min(u_{\theta'}, \theta')$
> **end for**
> **for all** $c \in \mathcal{C}$ **do**
>> { // perturb keeping the simplex restrictions }
>> $\pi'_c \leftarrow \pi_c + \epsilon$
>> $P'_c \leftarrow P_c + \epsilon$
> **end for**
> { // perturb keeping the simplex restrictions }
> $\pi'_{\mathcal{C}} \leftarrow \pi_{\mathcal{C}} + \epsilon$
> $P'_{\mathcal{C}} \leftarrow P_{\mathcal{C}} + \epsilon$

---

# NUMERICAL EXPERIMENTS

The results of the numerical experiments conducted were compared to state-of-the-art metaheuristics and another recent hyper-heuristic that incorporates several state-of-the-art metaheuristic operators.

## Benchmarks

For the numerical experiments, we used six well-known continuous benchmark functions: Rastrigin, Rosenbrock, Styblinski Tang, Schweffel 2.23, Trid, and Qing. The basic properties of the test functions are presented in Table 1. The dimensionality, convexity, separability, and multimodality of the functions were varied to assess performance in different settings.

## Parameter tuning

The performance of the simulated annealing algorithm within the hyperlevel is sensitive to the initial temperature. The performance of the process in the case of the Rosenbrock function was assessed in various dimensions with varying initial temperatures and iterations. Figure 2 depicts these results. The plots show that having a higher initial temperature (10,000) yields improved, more robust results. Detailed results are presented in the Supplemental Files.

---

**Algorithm 4** Base level optimization process

---

$\{ l_x, u_x$ - problem bounds $\}$

$\{ G$ - current generation of offspring $\}$

$\{ G'$ - next generation of offspring $\}$

$\{ best(G)$ - cost of best performing offspring in $G \}$

$\{ o_s^c$ - next operator state in the $c$ category$\}$

$\{ C_s$ - next category state$\}$

$\{ f_e$ - objective function evaluation count $\}$

$\{ f_o$ - function evaluations required by operator $o\}$

**for all** $x \in G$ **do**

$\quad x \leftarrow U(l_x, u_x)$ $\{//$ random uniform offspring$\}$

**end for**

$G' \leftarrow G$

**for all** $c \in \mathcal{C}$ **do**

$\quad o_s^c \leftarrow o, o \sim \pi_c$ $\{//$ inital operator state with $\pi_c$ distribution$\}$

**end for**

$\mathcal{C}_s \leftarrow c, c \sim \pi_{\mathcal{C}}$ $\{//$ inital category state with $\pi_{\mathcal{C}}$ distribution$\}$

$f_e \leftarrow 0$

**while** $f_e < h_f$ **do**

$\quad G, G' \leftarrow o_s^{\mathcal{C}_s}(\beta_{\mathcal{C}_s}, G, G')$ $\{//$ apply the next operator $\}$

$\quad f_e \leftarrow f_e + f_{o_s^{\mathcal{C}_s}}$ $\{//$ track function evaluations$\}$

$\quad \mathcal{C}_s \leftarrow c, c \sim P_{\mathcal{C}, \mathcal{C}_s}$ $\{//$go to next category state from $\mathcal{C}_s\}$

$\quad o_s^{\mathcal{C}_s} \leftarrow o, o \sim P_{\mathcal{C}_s, o_s^{\mathcal{C}_s}}$ $\{//$ go to next state in $P_{\mathcal{C}_s}$ from $o_s^{\mathcal{C}_s}\}$

**end while**

**return** $\min(best(G), best(G'))$ $\{//$ return the minimal cost$\}$

---

**Table 1 Test functions and their properties used for numerical experiments.**

| Function name | Properties |
|---|---|
| Qing | Non-convex, separable, multimodal |
| Rastrigin | Non-convex, separable, multimodal |
| Rosenbrock | Non-convex, non-separable, multimodal |
| Schweffel 2.23 | Convex, separable, unimodal |
| Styblinski Tang | Non-convex, separable, multimodal |
| Trid | Convex, non-separable, unimodal |

## Comparison with other methods

For comparisons, we use another recent hyper-heuristic, the CUSTOMHyS: Customising Optimisation Metaheuristics *via* Hyper-heuristic Search framework (downloaded from https://github.com/ElsevierSoftwareX/SOFTX-D-20-00035) (*Cruz-Duarte et al., 2020*).

CUSTOMHyS applies operators from several well known metaheuristics on the base level:

**Algorithm 5** Combined one point and arithmetic crossover for GA

$|G|$ { - number of offspring in $G$}
$n$ { - dimensionality of the objective function}
$\alpha$ { - arithmetic crossover coefficient}
$c_r$ { - crossover rate}
$m_r$ { - mutation rate}
$m_\sigma$ { - standard deviation of mutation distribution}
$cp_r$ { - crossover point ratio}
$p_r$ { - parent pool ratio}
$U(G,p)$ { = $\{x_i \in G : U(0,1) < p\}$-random subset of $G$}
**for** $i \leftarrow 0,\, i < |G|,\, i \leftarrow i+1$ **do**
  $x'_i \leftarrow x_i$ { // $x$ parent, $x'$ child}
  **if** $U(0,1) < c_r$ **then**
    $x_{j_1} \leftarrow$ best offspring in $U(G,p_r)$ such that $j_1 \neq i$
    $x_{j_2} \leftarrow$ best offspring in $U(G,p_r)$ such that $j_1 \neq j_2 \neq i$
    **for** $k \leftarrow 0,\, k < n,\, k \leftarrow k+1$ **do**
      **if** $k < n \cdot cp_r$ **then**
        $x'_{i,k} \leftarrow \alpha \cdot x_{j_1,k} + (1-\alpha) \cdot x_{j_2,k}$
      **else**
        $x'_{i,k} \leftarrow \alpha \cdot x_{j_2,k} + (1-\alpha) \cdot x_{j_1,k}$
      **end if**
      **if** $U(0,1) < m_r$ **then**
        $x'_{i,k} \leftarrow x_{i,k} + \epsilon,\, \epsilon \sim \mathcal{N}(0, m_\sigma)$
      **end if**
    **end for**
  **end if**
**end for**

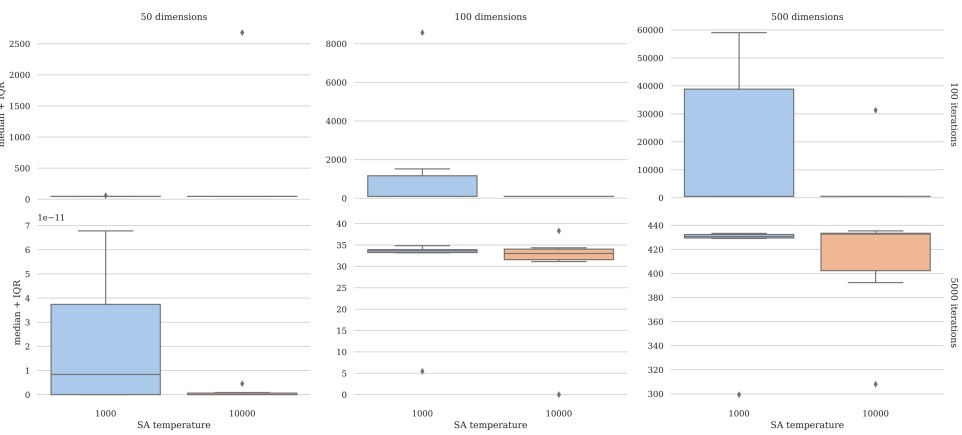

**Figure 2** **Parameter tuning results.** The higher initial temperature improves performance.

- it uses the central force dynamic operator of the central force optimisation (CFO) (*Formato, 2008*) algorithm;
- differential crossover and mutation operators from differential evolution (DE) (*Storn & Price, 1997*);
- the genetic mutation and crossover operators from genetic algorithm (GA) (*Whitley, 1994*);
- the spiral dynamic operator of stochastic spiral optimisation (SSO) (*Cruz-Duarte et al., 2017*);
- the gravitational search operator of the gravitational search algorithm (GSA) (*Rashedi, Nezamabadi-pour & Saryazdi, 2009*);
- the swarm dynamic operator from particle swarm optimisation (PSO) (*Kennedy & Eberhart, 1995*);
- the firefly dynamic operator from firefly algorithm (FA) (*Gandomi, Yang & Alavi, 2011*);
- the random flight and search operators from random search (RS) and
- uniform random sampling, and the local random walk operator from cuckoo search (CS) (*Yang & Deb, 2013*).

CUSTOMHyS uses simulated annealing on the hyperlevel and searches for a fixed-length operator sequence that is repeated.

The comparison with CUSTOMHyS was performed on the six benchmark functions, using the same number of function evaluations, base-level sample size, simulated annealing steps, population and problem size. The approaches were tested with each offspring being limited to 100 function evaluations to highlight the limitations that appear in costly optimization problems. The base-level performance sample size was fixed at 30 to ensure statistical significance. The number of simulated annealing steps was limited to 100. The population size was fixed at 30. Various problem dimensionalities were considered (5, 50, 100, 500). The minimal median plus interquartile range of the performances at each simulated annealing step was considered the final performance of each method. This metric was chosen so that the performances were not affected by outliers.

We compare our results with the following state-of-the-art metaheuristics:

1. The slime mould algorithm (SMA) (*Li et al., 2020*) which is a biology-inspired metaheuristic that is based on the oscillation of slime mould;
2. the artificial ecosystem optimizer (AEO) (*Zhao, Wang & Zhang, 2019*) which is a system-based heuristic that mimics the behavior of an ecosystem of living organisms;
3. the battle royal optimizer (BRO) (*Rahkar Farshi, 2020*) which is a human-based metaheuristic that simulates a survival game;
4. the Archimedes optimization algorithm (ArchOA) (*Hashim et al., 2020*) which is a physics-inspired metaheuristic that imitates the phenomenon of buoyancy of objects immersed in a fluid;
5. the particle swarm optimizer (PSO) (*Kennedy & Eberhart, 1995*) which is a swarm-based metaheuristic that simulates the movement of particles and
6. the coral reef optimizer (CRO) (*Salcedo-Sanz et al., 2014*) which is a nature-inspired algorithm that simulates the growth of coral reefs.

For the comparison, we used the implementations provided by the MEALPY (downloaded from https://github.com/thieu1995/mealpy, version 2.5.1) library (*Thieu & Mirjalili, 2022*), a software package containing most of the cutting-edge metaheuristic algorithms.

The six metaheuristics were tested in similar settings, resulting in the same number of total function evaluations. This was achieved by having equal numbers of performance samples and hyper-heuristic steps; that is, each performance sample had the same size and total function evaluations as in a hyper-heuristic step. All populations were initialized uniformly within the problem bounds and were of the same size.

The experiments were performed on a system equipped with an Intel Core i7-9750H CPU, NVIDIA GeForce GTX 1660 Ti Mobile GPU, and 16 GB of RAM running Ubuntu 20.04.1 LTS. The 11.4 version of the CUDA Toolkit was used along with Python version 3.8.10. The results of the experiments are available and can be reproduced with the public NMHH implementation, which can be accessed on GitHub (https://github.com/BNandor/MatOpt/tree/main/NMHH).

*Metaheuristic parameters.* The NMHH operator parameter bounds used are as follows: the DE operator force $F \in [0.4, 0.7]$ and crossover rate $c_r \in [0.9, 1]$, the GA operator one-point crossover rate and point $c_r, cp_r \in [0, 1]$, the arithmetic crossover constant $\alpha \in [0, 1]$; the mutation rate and size $m_r \in [0, 0.1], m_\sigma \in [0, 100]$, and the ratio of the parent pool $p_r \in [0.2, 1]$. The GD and LBFGS initial step lengths were $\alpha \in [0.5, 5]$. The evaluation limits were set to $f_{GD} \in [1, 3], f_{LBFGS} \in [6, 10]$. The LBFGS memory was fixed to 5, and the step coefficients were $c_1 \in [0, 0.1], c_2 \in [0.8, 1]$.

The parameters of the state-of-the-art metaheuristics were set to those suggested by the MEALPY package. For the CRO, the rate of occupation was set to 0.4, the broadcast/existing rate ($F_b$) to 0.9, the duplication rate ($F_a$) to 0.1, the depredation rate ($F_d$) to 0.1, the maximum depredation probability ($P_d$) to 0.5, the probability of the mutation process ($GCR$) to 0.1, the mutation process factors $gamma_{min}$ to 0.02, $gamma_{max}$ to 0.2, and the number of attempts of a larva to set in reef ($n_{trials}$) to 5. For BRO, the dead threshold was set to 3. For the ArchOA, the factors were set to $c_1 = 2, c_2 = 5, c_3 = 2$ and $c_4 = 0.5$. The accelerations were set to $acc_{min} = 0.1$ and $acc_{max} = 0.9$. The AEO does not expose any parameters. The SMA probability threshold was set to 0.3. The local and global coefficient of the PSO was set to 2.05, the minimum weight to 0.4 and the maximum weight to 0.9.

For CUSTOMHyS, the suggested heuristic collection and parameters were used: The simulated annealing initial temperature (*max_temperature*) was set to 200, the temperature cooling rate (*cooling_rate*) to 0.05, the stagnation rate (*stagnation_percentage*) to 0.3 and the length of the operator sequence (*cardinality*) to 3. The population size (*num_agents*) was set to 30, and each offspring was limited to 100 iterations (*num_iterations*). The number of hyper-heuristic steps (*num_steps*) was limited to 100 with each step having a performance sample size (*num_replicas*) of 30. The suggested heuristic (*default*) collection contains variations of twelve operators: random search, central force dynamic, differential mutation, firefly dynamic, genetic crossover, genetic mutation, gravitational search, random flight, local random walk, random sample, spiral dynamic and swarm dynamic.

**Table 2 Results obtained for the six test functions.** The minimal median plus interquartile range is presented. Best ranking results according to the Wilcoxon rank-sum test are highlighted in bold.

| problem | dimension | NMHH | CUSTOMHyS | SMA | AEO | BRO | ArchOA | PSO | CRO |
|---|---|---|---|---|---|---|---|---|---|
| Qing | 5 | **1.3805e−30** | 6.9269e−27 | 7.7565e−02 | 6.6547e−01 | 1.2591e+01 | 1.9785e+01 | 1.0583e+03 | 1.0405e+05 |
| | 50 | **3.8581e−03** | 3.5255e+05 | 1.4020e+04 | 2.0684e+04 | 2.5249e+04 | 3.8629e+04 | 2.8970e+10 | 1.2337e+11 |
| | 100 | **2.1094e−05** | 1.0445e+08 | 1.6719e+05 | 1.8468e+05 | 2.0831e+05 | 3.0352e+05 | 7.2997e+10 | 4.4440e+11 |
| | 500 | **5.5487e−01** | 2.7925e+08 | 3.2776e+07 | 2.5715e+07 | 2.5508e+07 | 3.2362e+07 | 5.1137e+12 | 4.2136e+12 |
| Rastrigin | 5 | 6.8168e−03 | 2.0791e+00 | **0.0000e+00** | **0.0000e+00** | **0.0000e+00** | 5.3705e+00 | 2.2249e+01 | 1.1563e+01 |
| | 50 | 8.1054e+01 | 1.2439e+02 | **0.0000e+00** | **0.0000e+00** | **0.0000e+00** | 2.1727e+02 | 6.1434e+02 | 4.1291e+02 |
| | 100 | 2.5358e+02 | 3.3217e+02 | **0.0000e+00** | **0.0000e+00** | **0.0000e+00** | 2.7143e+00 | 1.2222e+03 | 1.0735e+03 |
| | 500 | 2.1600e+03 | 5.4562e+03 | **0.0000e+00** | **0.0000e+00** | **0.0000e+00** | 1.1132e+00 | 8.6621e+03 | 7.4188e+03 |
| Rosenbrock | 5 | **1.8418e+00** | 3.9387e+00 | 3.3818e+00 | 3.8733e+00 | 4.0155e+00 | 4.1998e+00 | 2.9554e+02 | 1.0212e+03 |
| | 50 | **4.5794e+01** | 4.9130e+03 | 4.8965e+01 | 4.8951e+01 | 4.8649e+01 | 4.9374e+01 | 4.1971e+07 | 1.4990e+08 |
| | 100 | **9.5543e+01** | 1.5961e+05 | 9.8964e+01 | 9.8954e+01 | 9.8281e+01 | 9.9508e+01 | 9.5397e+07 | 5.6190e+08 |
| | 500 | **4.9184e+02** | 1.4714e+07 | 4.9896e+02 | 4.9896e+02 | 4.9535e+02 | 4.9962e+02 | 6.6368e+09 | 5.4849e+09 |
| Schwefel223 | 5 | 9.5097e−17 | 4.5248e−92 | **0.0000e+00** | **0.0000e+00** | **0.0000e+00** | 3.0466e−26 | 5.7078e−09 | 4.3097e−04 |
| | 50 | 8.0301e−05 | 4.5912e−04 | **0.0000e+00** | **0.0000e+00** | **0.0000e+00** | 1.8737e−18 | 1.6220e+09 | 1.2841e+09 |
| | 100 | 7.7415e−05 | 1.6397e+02 | **0.0000e+00** | **0.0000e+00** | **0.0000e+00** | 5.6318e−18 | 4.4789e+09 | 8.9519e+09 |
| | 500 | 3.4822e−04 | 9.4162e+08 | **0.0000e+00** | **0.0000e+00** | **0.0000e+00** | 3.9716e−18 | 2.8568e+11 | 2.0302e+11 |
| Styblinskitang | 5 | **−1.9583e+02** | **−1.9583e+02** | −1.9569e+02 | −1.8076e+02 | −1.6963e+02 | −1.7348e+02 | −1.7592e+02 | −1.8158e+02 |
| | 50 | **−1.9018e+03** | −1.6965e+03 | −1.3724e+03 | −1.0942e+03 | −1.5435e+03 | −1.0184e+03 | −8.9307e+02 | −1.2202e+03 |
| | 100 | **−3.4077e+03** | −2.8352e+03 | −2.3018e+03 | −2.0178e+03 | −3.0103e+03 | −1.8841e+03 | −1.5568e+03 | −2.0718e+03 |
| | 500 | **−1.6526e+04** | −8.4084e+03 | −7.9417e+03 | −9.0582e+03 | −1.4859e+04 | −8.7477e+03 | −4.1091e+03 | −6.3357e+03 |
| Trid | 5 | **−3.0000e+01** | **−3.0000e+01** | −2.9890e+01 | −2.9916e+01 | −2.4194e+01 | −2.8117e+01 | −2.9704e+01 | −2.1214e+01 |
| | 50 | **−8.1106e+03** | 2.5636e+05 | 2.6391e+01 | 4.4701e+01 | −1.0558e+02 | 8.5830e+01 | 1.6161e+07 | 2.8479e+07 |
| | 100 | **−4.1094e+04** | 2.9695e+07 | 7.9457e+01 | 9.5869e+01 | −1.6720e+02 | 7.4611e+02 | 5.6313e+08 | 1.3875e+09 |
| | 500 | **−2.1395e+05** | 6.2820e+11 | 4.8293e+02 | 4.9688e+02 | −8.6542e+02 | 6.5888e+05 | 9.0493e+12 | 7.4315e+12 |

The parameters of all 205 variations can be found in the CUSTOMHyS implementation (https://github.com/ElsevierSoftwareX/SOFTX-D-20-00035, last accessed 12/11/2022) (*Cruz-Duarte et al., 2020*).

## Results and discussion

Numerical results are presented in Table 2. The best results are highlighted in bold, and the Wilcoxon rank-sum statistical test was used for comparison. The results point to a considerable difference between the hyper-heuristic and standard metaheuristic approaches.

For the majority of problems (Qing, Rosenbrock, Styblinski Tang, Trid), NMHH outperformed all metaheuristics and CUSTOMHyS. In the case of Rastrigin and Schwefel 2.23, the SMA, AEO and BRO found the global minimum. For Rastrigin, both hyper-heuristic approaches performed considerably worse than the metaheuristics, but for Schwefel 2.23, the NMHH approximated the solution several magnitudes better than CUSTOMHyS. NMHH found the best solution in 66% of test cases. These results indicate that in many cases our approach can outperform state-of-the-art metaheuristics and the investigated selection hyper-heuristic.

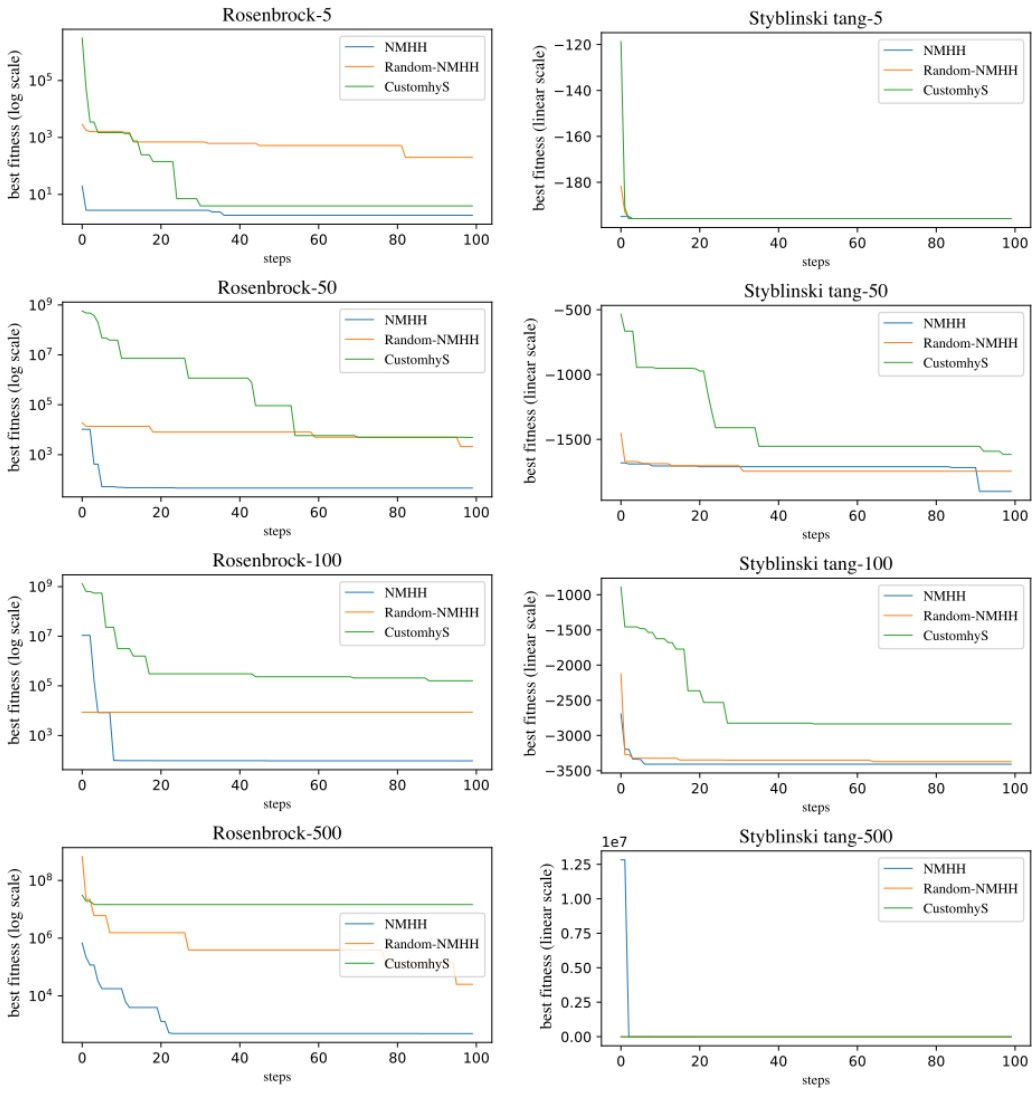

**Figure 3** **Evolution of the Rosenbrock, Styblinski Tang problems over time, comparing the CUSTOMHyS and two variants of the method: the presented SA based (NMHH) and the random based variant (Random NMHH).** The evolution of the minimum median plus interquantile range is shown.

*Convergence.* Figure 3 presents the evolution of the performances for the Rosenbrock and Styblinski Tang functions. The plots highlight the ability of NMHH to find the best-performing operators and their sequences. We compared NMHH and CUSTOMHyS to a modified version of the proposed method (Random NMHH), where simulated annealing was replaced with random uniform selection and generation. Random NMHH performed worse than the simulated annealing variant in all cases, pointing to the importance of the selection and generation mechanism in the hyperlevel.

**Table 3** **Average and standard deviation of the computational times measured in seconds for the Rosenbrock function.** Best results are highlighted in bold.

| Dimension | NMHH (sec) | CUSTOMHyS (sec) |
|---|---|---|
| 5 | **373.31 ± 44.20** | 682.55 ± 186.73 |
| 100 | **540.7 ± 57.96** | 4771.14 ± 4578.12 |

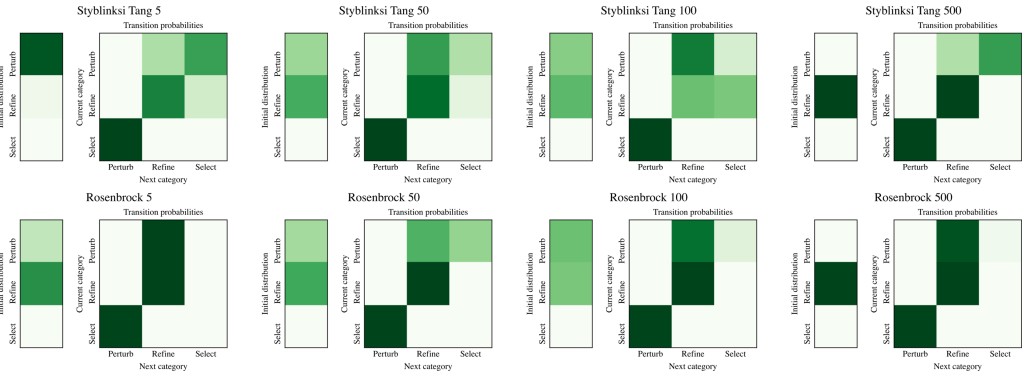

**Figure 4** **The best performing operator category transition matrices for Styblinksi Tang and Rosenbrock functions.** For Styblinksi Tang the alternation of perturb and selection operators with the occasional refinement performed best. In the case of the Rosenbrock function the evolved sequences initially perform perturbation and finish with constant refinement.

*Computational time.* Table 3 shows the measured computational times for the Rosenbrock function in low- and high-dimensional settings for 10 runs. In the low-dimensional setting, both NMHH and CUSTOMHyS had times of the same order of magnitude, but in the high-dimensional setting, NMHH performs an order of magnitude better than CUSTOMHyS and has a lower variance. This shows that the NMHH implementation scales well with increasing dimensionality.

*Evolved Markov chain.* The ability of NMHH to adapt the used operator and operator category distributions to the problem is best seen in Figs. 4 and 5. Figure 4 depicts the best-performing operator category transition matrices and initial distributions of the upper layer in the case of Styblinski Tang and the Rosenbrock function. Figure 5 depicts the best-performing operator transitions. NMHH adapted the exploration-exploitation rate to the shape of the cost landscapes. The shape of Styblinski Tang favours a balance between exploration and exploitation as it contains many local minima. The transition matrices reflect that the best-performing distributions include both iterated local search and perturb operators. In 500 dimensions, the optimal optimization approach proved to be more balanced towards continuous refinement. NMHH adapted to the valley-shaped landscape of the Rosenbrock test function and evolved to use the iterated local search approach in the limited function evaluation setting.

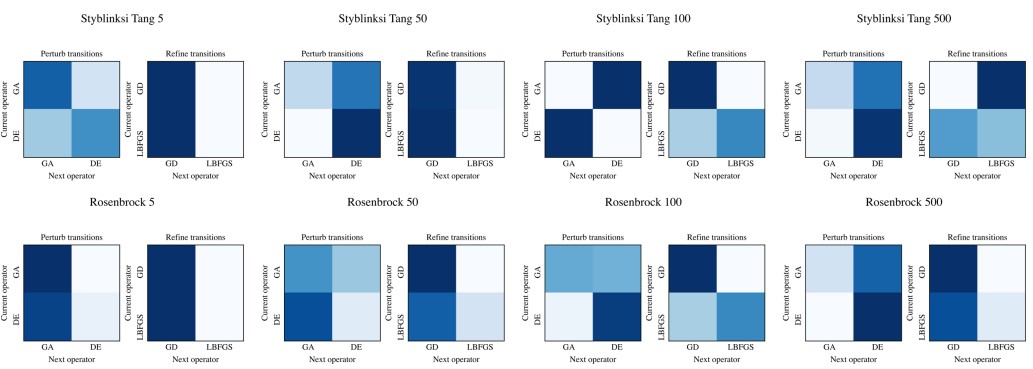

**Figure 5** **The best performing operator transition matrices for Styblinksi Tang and Rosenbrock functions.** While the perturbating operators alternate in the majority of cases, for refinement the gradient descent operator was preferred.

**Table 4** Evolved operator sequences of CUSTOMHyS for Rosenbrock and Styblinksi Tang.

| Dimension | Rosenbrock | Styblinski Tang |
|---|---|---|
| 5 | Genetic crossover$_{120}$, swarm dynamic$_{194}$, differential mutation$_{21}$ | Random search$_{171}$, random flight$_{139}$ |
| 50 | Gravitational search$_{135}$, swarm dynamic$_{191}$, genetic crossover$_{115}$ | Genetic crossover$_{73}$, spiral dynamic$_{180}$, random search$_{172}$ |
| 100 | Genetic crossover$_{117}$, swarm dynamic$_{203}$, random search$_{169}$ | Differential mutation$_{19}$, swarm dynamic$_{193}$, genetic crossover$_{56}$ |
| 500 | Random search$_{168}$, local random walk$_{158}$, genetic crossover$_{11}$ | Genetic crossover$_{99}$, random search$_{174}$, genetic crossover$_{98}$ |

*Evolved CUSTOMHyS operator sequences.* Table 4 depicts the operator sequences that CUSTOMHyS evolved for Rosenbrock and Styblinski Tang and shows the index of the heuristic operator variation within the heuristic collection.

# CONCLUSION AND FURTHER WORK

Optimization plays an important role in computational tasks. Hyper-heuristics are a new paradigm for solving optimization problems as they can significantly improve numerical results. In this article, we propose a new hyper-heuristic framework (NMHH) with two main innovations: the use of a nested Markov chain to model complex distributions of operators and the search for the equilibrium between exploration and exploitation in this space by balancing the category of iterated local search against metaheuristic exploratory operators. Numerical experiments conducted on continuous benchmark problems in high dimensions confirm the effectiveness of the proposed approach. The results show that NMHH evolved operator sequences and found the exploration-exploitation rate that outperformed state-of-the-art metaheuristics and the CUSTOMHyS hyper-heuristic in 66% of the cases in the high-dimensional setting.

As further work, other metaheuristics can be introduced at the base level. Detecting point clusters that are converging to the same fixed point and keeping the best one while perturbing the others could facilitate the better exploration of the attraction basins, improving performance. Another research direction could be the hybridization of the simulated annealing search operator within the hyperlevel with local search operators. The formulation of the change of operator parameters during the optimization process as an optimal control problem is another interesting research direction.

### Funding
This work was supported by a grant of the Ministry of Research, Innovation and Digitization, CNCS—UEFISCDI, project number PN-III-P1-1.1-TE-2021-1374, within PNCDI III. The funders had no role in study design, data collection and analysis, decision to publish, or preparation of the manuscript.

### Grant Disclosures
The following grant information was disclosed by the authors:
The Ministry of Research, Innovation and Digitization, CNCS—UEFISCDI, project number PN-III-P1-1.1-TE-2021-1374, within PNCDI III.

### Competing Interests
The authors declare there are no competing interests.

### Author Contributions
- Nándor Bándi conceived and designed the experiments, performed the experiments, analyzed the data, performed the computation work, prepared figures and/or tables, and approved the final draft.
- Noémi Gaskó conceived and designed the experiments, analyzed the data, prepared figures and/or tables, authored or reviewed drafts of the article, and approved the final draft.

### Data Availability
The data is available at GitHub and Zenodo:

- https://github.com/BNandor/MatOpt/tree/main/NMHH

- Bandi Nandor. (2023). BNandor/MatOpt: initial release (v1.0.0). Zenodo. https://doi.org/10.5281/zenodo.10201026.

### Supplemental Information
Supplemental information for this article can be found online at http://dx.doi.org/10.7717/peerj-cs.1785#supplemental-information.

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
