# Peer review of "Nested Markov chain hyper-heuristic (NMHH): a hybrid hyper-heuristic framework for single-objective continuous problems"

_PeerJ Computer Science, doi:10.7717/peerj-cs.1785_

## Round 0.1 · original submission · Major Revisions

Dear authors,

Your article has not been recommended for publication in its current form. However, we encourage you to address the concerns and criticisms of the reviewers, particularly in terms of readability, validity of the findings, and general quality, and to resubmit your article once you have updated it accordingly.

The following points should also be addressed:

1- The research gaps and contributions should be clearly summarized in the introduction section. Please evaluate how your study is different from others in the related work section.
2- Please include future research directions.
3- All of the values of the algorithm parameters selected for comparison should be listed.
4- The paper lacks the running environment, including software and hardware. The analysis and configurations of experiments should be presented in detail for reproducibility. It is convenient for other researchers to redo your experiments and this makes your work easy acceptance.
5- The authors should clarify the pros and cons of the methods. What are the limitation(s) methodology(ies) adopted in this work? Please indicate practical advantages, and discuss research limitations.

Reviewer 2 has requested that you cite specific references. You are welcome to add it/them if you believe they are relevant. However, you are not required to include these citations, and if you do not include them, this will not influence my decision.


Best wishes,

**Language Note:** The Academic Editor has identified that the English language must be improved. PeerJ can provide language editing services - please contact us at [email protected] for pricing (be sure to provide your manuscript number and title). Alternatively, you should make your own arrangements to improve the language quality and provide details in your response letter. – PeerJ Staff

·

Basic reporting

[YES] Clear, unambiguous, professional English language is used throughout:
No comment.

[YES] Intro & background to show context. Literature well referenced & relevant:
It is necessary to give a more harmonious sequence to the introduction. In addition, it is essential to include critical references, for example:
- Line 25: "Some nature-inspired, some physics-inspired, iterative, and hybrid."
- Line 26: "Finding the appropriate approach is often a specific problem and can be tedious."
- Line 28: "Iterative methods converge faster to a local minimum, but are highly dependent on the initial solution."

[YES] Professional article structure, figures, tables. Raw data shared.
No comment.

[YES] Figures are relevant, high quality, well labelled & described:
They are acceptable but are not very descriptive. For example, Figures 2 and 3 must be improved.

Experimental design

[YES] Original primary research within the Scope of the journal:
No comment.

[NO] Research question well defined, relevant & meaningful. It is stated how the research fills an identified knowledge gap:
The Research Question is not clear. It is necessary to dig a little into the document to understand it.

[NO] Rigorous investigation performed to a high technical & ethical standard:
Many issues need to be fixed. I strongly recommend the authors to revise carefully their report.

[NO] Methods described with sufficient detail & information to replicate:
No, it is necessary to include more information about the reference framework designed, such as the operators used, i.e. to define the perturbator mathematically. It is essential to detail the formalism of the proposal. In addition, free access to the designed framework is required to interact with the proposal and facilitate the replication of this work, which will benefit the journal and the article's citation.

Validity of the findings

[NO] Impact and novelty not assessed. Meaningful replication encouraged where rationale & benefit to literature is clearly stated:
The authors must provide free access to the designed framework and the resulting dataset to allow readers to interact with the proposal and facilitate the replication of this work, which will benefit the journal and, obviously, the article's cites.

[NO] All underlying data have been provided; they are robust, statistically sound, & controlled:
No, it is necessary to give us the resulting numerical data. In addition, the statistical tests do not appear in the document. I suggest avoiding long tables and perhaps using other resources to analyse data, such as boxplots, statistics, etc.

[YES] Conclusions are well stated, linked to original research question & limited to supporting results:
No comment.

Additional comments

General Summary:

This paper introduces a perspective within the domain of Hyper-heuristics by proposing a hybrid approach that employs a nested Markov chain structure intending to address single-objective continuous optimisation problems. This approach uses a Markov chain at the base level to systematically search for operators and sequences that maximise performance. These operators are composed of a genetic algorithm and differential evolution parts. In addition, the parameter tuning of each operator was performed to maximise the overall algorithm performance. Simultaneously, at the top level of this structure, the metaheuristic technique known as Simulated Annealing was implemented to evolve the Markov chain and dynamically adjust the parameters of the operators. To validate this proposal's effectiveness, the results from the experiments were compared with those of six metaheuristics recognised in the field, and a comparison was made with an established framework.


Strengths of the document:

The proposal is interesting. The presented HH model can select and tune operators to maximise the performance of the heuristic-based procedure. Moreover, including the nested Markov chain provides a crucial formalism (a fact that needs to be exploited) for the generated sequence.


Weaknesses of the paper:

- Robust and transparent statistical tests are needed. I suggest the implementation of a Wilcoxon test and thus really verify the performance of the proposed algorithms.
- It is vital to show the computational cost of the proposal and compare it with the selected literature proposals.
- It is necessary to include the hyper-heuristic evolution process until obtaining the final sequence.
- It is neither shown nor described which sequence was generated or adjusted parameters.
- The operator generated by CUSTOMHYS was not shown. Did you perform several repetitions of it?
- The comparison is a little unfair because the other methods with which it is compared do not present an adjustment of their hyperparameters. Plus, some of these render a heuristic-based algorithm instead of a solution. So, it seems the authors are comparing hyper-specialised methods against others which may take some time to find a good solution.
- Your proposal cannot be directly compared with the SAHH or RSHH implemented in CUSTOMHYS because they do not build the solution online. In my opinion, the proposed method is not a hyper-heuristic, at least not in the same context as SAHH or RSHH; in that sense, it is a bare hybrid metaheuristic.
- The numerical results could be presented in box or violin plots.
- It is not specified in which scenario the metaheuristic was generated, neither for the proposal nor for CUSTOMHYS.

Reviewer 2 ·

Basic reporting

A new hyper-heuristic framework (NMHH) is proposed. It employs a nested Markov chain to model complex distributions of operators. It also balances exploration and exploitation aiming at equilibrium.

Nevertheless, some points that the authors should consider are:

ITEM 1) The section entitled “RELATED WORK” is incomplete because nothing is mentioned about how the hyperparameters need to be adjusted in order to speed-up the search. In the literature, there is a wide variety of methods for parameter selection that should be covered in an extra paragraph. A reference to the following paper, where the agents’ parameters and operators are adaptively chosen on-line, should be included:
Oteiza P.P., Ardenghi J.I., Brignole N.B., “Parallel Hyper-heuristics For Process Engineering Optimization” Special Issue of Comp. & Chem. Eng. (ISSN: 0098-1354) https://doi.org/10.1016/j.compchemeng.2021.107440 , 153, 107440, 2021

ITEM 2) Table 2 is too long. It should be replaced by representative plots. Besides, the long list should also be presented as supplementary material, which may be useful for some readers.

ITEM 3) The writing style should be improved to make the article easier to read. Please make the following changes:
A) There is a very long sentence ranging from lines 151 to 161. It is difficult to read- Moreover, it has some punctuation mistakes. Use a comma before which and separate the ítems with semi-colons.Then, the enumeration should be arranged by modifying the text in this way:
“We compare our results with the following state-of-the-art metaheuristics:
1. the Coral Reef Optimizer (CRO) (Salcedo-Sanz et al., 2014), which is a nature-inspired algorithm that simulates the growth of coral reefs;
2. the Particle Swarm Optimizer (PSO) (Kennedy and Eberhart, 1995), which is a swarm-based metaheuristic that simulates the movement of particles;
3. the Artificial Ecosystem Optimizer (AEO) (Zhao et al., 2019), which is a system-based heuristic that mimics the behavior of an ecosystem of living organisms;
4. the Battle Royal156 Optimizer (BRO) (Rahkar Farshi, 2020), which is a human-based metaheuristic that simulates a survival game;
5. the Slime Mould Algorithm (SMA) (Li et al., 2020), which is a biology-inspired metaheuristic that is based on the oscillation of slime mould and
6. the Archimedes Optimization algorithm (ArchOA) (Hashim et al., 2020), which is a physics-inspired metaheuristic that imitates the phenomenon of buoyancy of objects immersed in a fluid. “

B) There is another sentence that is too long. It ranges from lines 164 to 173. To improve readability, make similar changes to the ones suggested above.

Experimental design

Numerical experiments were conducted on six benchmark problems for various dimensions in order to confirm the effectiveness of the proposed approach. The choice of these continuous benchmark functions is adequate and challenging because they are well-known for their intrinsic difficulties related to optimization.

Validity of the findings

No comment.

Additional comments

No comment

---

## Round 0.2 · accepted · Accept

Dear authors,

Thank you for the revision and for clearly addressing all the reviewers' comments. I confirm that the paper is improved and addresses the concerns of the reviewers. Your paper is now acceptable for publication in light of this revision.

Best wishes,